# Investigating the Impacts of Diet, Supplementation, Microbiota, Gut–Brain Axis on Schizophrenia: A Narrative Review

**DOI:** 10.3390/nu16142228

**Published:** 2024-07-11

**Authors:** Izabela Zajkowska, Patrycja Niczyporuk, Agata Urbaniak, Natalia Tomaszek, Stefan Modzelewski, Napoleon Waszkiewicz

**Affiliations:** Department of Psychiatry, Medical University of Bialystok, pl. Wołodyjowskiego 2, 15-272 Białystok, Poland; 36831@student.umb.edu.pl (I.Z.); napoleon.waszkiewicz@umb.edu.pl (N.W.)

**Keywords:** gut–brain axis, diet, schizophrenia, supplementation, microbiota, eating habits, neuroleptics

## Abstract

Schizophrenia is a disease with a complex etiology that significantly impairs the functioning of patients. In recent years, there has been increasing focus on the importance of the gut microbiota in the context of the gut–brain axis. In our study, we analyzed data on the gut–brain axis in relation to schizophrenia, as well as the impacts of eating habits, the use of various supplements, and diets on schizophrenia. Additionally, the study investigated the impact of antipsychotics on the development of metabolic disorders, such as diabetes, dyslipidemia, and obesity. There may be significant clinical benefits to be gained from therapies supported by supplements such as omega-3 fatty acids, B vitamins, and probiotics. The results suggest the need for a holistic approach to the treatment of schizophrenia, incorporating both drug therapy and dietary interventions.

## 1. Introduction

Schizophrenia is a mental disorder with a complex etiology. It affects approximately 1 percent of the world’s population [1]. The life expectancy of people with schizophrenia is 10–20 years shorter than that of the general population, which is associated with a number of risk factors that are more common in this group, such as obesity, diabetes, and lipid abnormalities. The lifestyle of schizophrenic patients, characterized by less physical activity and a poor diet, also contributes to increased metabolic risk [2]. The side effects of neuroleptics, which are known to cause weight gain and affect glucose and lipid metabolism, may be partly responsible for the increased prevalence of these health problems [3].

In recent years, there has been a growing interest to understand the gut–brain axis and its impact on mental health, including diseases such as schizophrenia. 

The gut–brain axis may be the pathway through which diet and supplementation affect disease. This is consistent with hypotheses that early-life *Helicobacter pylori* infection may lead to systemic infection, causing dopaminergic dysfunction, initiating inflammation, and altering cysteine production, increasing schizophrenia risk in genetically susceptible individuals [4,5]. These effects may be important for mood and affective disorders as well as for the negative symptoms of schizophrenia [6].

Our narrative review is motivated by the need to synthesize current research findings on the influences of diet, supplementation, and gut microbiota on schizophrenia. Because of the complexity of schizophrenia and the different methods that must be used for treating it, our review aims to provide a comprehensive overview that can aid in developing more effective and holistic therapeutic strategies. By integrating the findings from various studies, we seek to highlight potential dietary and supplementary interventions that could complement conventional pharmacotherapy and improve patient outcomes.

In our review, we will address several key points:Determine the role of the gut–brain axis in relation to schizophrenia (Section 3).Evaluate the importance of diet, supplementation, and selected common factors, like nicotinism, in relation to schizophrenia (Section 4, Section 5 and Section 6).Assess the impact of neuroleptic drugs on metabolic health (Section 7).

## 2. Materials and Methods

The PubMed, Google Scholar, and Web of Science databases were searched using the following key words: “gut–brain axis”, “diet”, “schizophrenia”, “supplementation”, “microbiota”, “eating habits”, “probiotics”, “prebiotics”, “dysbiosis”, “neuroleptics”, “treatment”, “metabolic syndrome”, “hyperlipidemia”, “hyperglycemia”, as well as combinations of these terms. We included relevant articles to assess the potential roles of microbiota, gut–brain axis, diets, and supplementation in relation to schizophrenia. It is essential to note that our review is not systematic, and despite attempts to cover all studies, one should keep in mind significant limitations.

## 3. The Gut–Brain Axis in the Pathogenesis of Schizophrenia

Emerging evidence underscores the role of the microbiota–gut–brain axis in the development of schizophrenia (SCZ) and other psychiatric disorders [7]. This bidirectional communication system between the gastrointestinal (GI) tract and the central nervous system (CNS) operates through neural, enteroendocrine, and immunological pathways [8]. Disruptions in the microbiota–gut–brain axis are implicated in the pathogenesis of SCZ, primarily involving inflammation, oxidative stress, tryptophan metabolism, mitochondrial dysfunction, as well as alterations in neurotransmitter and neurotrophic factor levels [9]. 

### 3.1. The Neural Pathway

The neural connection occurs between the central nervous system (CNS) and the enteric nervous system (ENS), a complex network of neurons abundant in small ganglia, and submucosal and myenteric neuronal plexuses [10]. Products of bacterial metabolism, gut hormones, and nutrients transmit signals to the CNS via the afferent branches of the vagus nerve. Conversely, efferent information is transferred from the CNS to regulate smooth muscle function, secretion, absorption, and intestinal blood flow [11]. The significance of the vagus nerve in this process was demonstrated in animal studies, where the administration of *Lactobacillus rhamnosus* induced alterations in γ-aminobutyric acid (GABA) receptor expression in mice, resulting in a decreased stress-induced corticosterone response and a reduction in anxiety and depressive-related behaviors. Notably, this effect was absent following vagotomy, suggesting the contribution of the vagus nerve in mediating these responses [12]. 

### 3.2. Neurotransmitters

The gut microbiome is capable of neurotransmitter production and takes part in transferring information. For example, *Lactobacillus* and *Bifidobacterium* produce gamma-aminobutyric acid (GABA) and acetylcholine, which act on receptors in the CNS [13]. *Escherichia* spp., *Candida* spp., and *Enterococcus* spp. can produce serotonin, *Bacillus* spp. dopamine, *Bacillus* spp., and *Saccharomyces* spp. noradrenaline [14]. Neurotransmitters synthesized by bacteria and enteroendocrine cells can enter the bloodstream and reach various parts of the body. Certain neurotransmitter precursors can penetrate the blood–brain barrier, contributing to the synthesis of neurotransmitters inside the brain. Moreover, gut sensory epithelial cells facilitate the rapid transmission of signals via the vagus nerve within milliseconds, achieved through the synthesis and release of neurotransmitters, such as glutamate [15]. What is more, certain Clostridium species have been found to inhibit the secretion of dopamine beta-hydroxylase, an enzyme responsible for converting dopamine into norepinephrine. Consequently, reduced levels of this enzyme lead to decreased norepinephrine levels and an accumulation of dopamine [16]. This disruption in the dopamine–norepinephrine balance is characteristic not only of schizophrenia, but also of compulsive behaviors and autism spectrum disorder [17]. 

Decreased levels of gamma-aminobutyric acid (GABA), an inhibitory neurotransmitter, are believed to potentially contribute to the pathogenesis of schizophrenia (SCZ) [18]. Postmortem analysis of individuals diagnosed with SCZ has revealed decreased concentrations of this neurotransmitter [19]. To further investigate this hypothesis, Ahn et al. conducted a study using iomazenil, an antagonist and partial inverse agonist of the benzodiazepine receptor, to induce a transient GABA deficit. The researchers observed exacerbated psychotic symptoms and perceptual difficulties among SCZ patients following iomazenil administration, but not among healthy controls [20]. 

Serotonin (5-HT, 5-hydroxytryptamine) is synthesized in various tissues, including the digestive tract, nervous system, and immune system, through the metabolism of tryptophan. Enterochromaffin cells (ECCs) are responsible for producing up to 95% of serotonin in the intestinal mucosa, with contributions also from the gut microbiota, neuronal plexuses, and muscular layers of the intestine [7]. Serotonin serves as a neurotransmitter involved in a range of neuropsychological processes, including mood regulation and pain perception, as well as in transmitting information within the enteric nervous system (ENS) and regulating gastrointestinal function [21]. One hypothesis suggests that stress-induced serotonin overload can lead to disrupted neuronal activity in the cerebral cortex. Moreover, both postmortem brain studies and in vivo investigations have revealed alterations in serotonin receptors (5-HTRs) and serotonin reuptake transporters (SERTs) in individuals diagnosed with schizophrenia (SCZ), suggesting a potential role for serotonin dysregulation in the pathophysiology of the disorder [22]. 

### 3.3. HPA Axis

Another pathway of communication between the CNS and gut microbiota is through the hypothalamic–pituitary–adrenal (HPA) axis, a neuroendocrine circuit crucial for the body’s response to stress stimuli [23]. The hypothalamus releases corticotropin-releasing hormone (CRH), stimulating the anterior pituitary gland to secrete the adrenocorticotropic hormone (ACTH). With the bloodstream, the ACTH reaches the adrenal cortex, prompting the secretion of glucocorticoids (GKSs), primarily cortisol [24]. 

Both elevated and reduced GKS levels can disrupt the delicate balance necessary for optimal brain development, as their receptors are distributed throughout the brain [25]. Dysregulation of the HPA axis might be a consequence of psychosocial stress and traumatic life events. Although the direct mechanisms are still unclear, it is noteworthy that past trauma increases the risk for psychosis, as well as the onset or exacerbation of schizophrenia in susceptible individuals [26,27]. Meta-analyses suggest that patients at high risk of psychosis tend to exhibit pituitary gland enlargement compared to healthy controls [28,29]. SCZ-diagnosed patients are also reported to have increased peripheral levels of morning cortisol compared to healthy controls. However, the quality of evidence supporting this finding is considered moderate [30]. Increased concentrations of cortisol are believed to take part in the onset of psychiatric disorders, especially during neurodevelopment [31]. Additionally, HPA axis hyperactivity might result in enhanced dopaminergic transmission, which is often referred to in the context of schizophrenia development.

In a study by Schmidt et al., the salivary cortisol levels of forty-five volunteers with no previous or current neurological or psychiatric disorders were measured. In order to standardize the intestinal environment of the volunteers, additional exclusion criteria were: the use of antibiotics, pre- or probiotics 3 months prior to the start of the study, and a vegan diet. Participants were assigned to a group receiving one of two prebiotics (fructooligosaccharides or galactooligosaccharides) or a placebo. Cortisol levels were found to be significantly lower after taking galactooligosaccharides compared to the placebo group [32].

A well-balanced microbiota contributes to maintaining the integrity of the intestinal barrier, particularly by preserving tight junctions between cells [33]. Acute stress, mediated by GKS, transiently alters gut motility and secretion, while chronic stress may lead to changes in gut microbiota composition, impact immune response, and alter intestinal barrier permeability. On the other hand, stress-inducing factors, such as dysbiosis, can over-activate and dysregulate HPA axis function [34]. The gut microbiota can activate the HPA axis via microbial antigens, cytokines, and prostaglandins that traverse the blood–brain barrier. Additionally, emerging evidence suggests that certain microbes can influence ileal corticosterone production and, by that, the activity of the HPA axis [26]. 

On the contrary, animal studies conducted on germ-free mice indicate that microbiome deficiency during brain development leads to elevated levels of cortisol and reduced levels of brain-derived neurotrophic factor (BDNF). Moreover, reconstitution with *Bifidobacterium infantis* reversed the excessive HPA stress response in mice [35]. Therefore, disturbances in the HPA axis and microbiota composition may hypothetically contribute to the pathogenesis of the disease.

### 3.4. Immunological Function

Exposing the immune system to a balanced microbiota initiates a process of specific immunological training. This training equips regulatory T cells to sustain a delicate balance between protecting against externally derived pathogens and tolerating commensal organisms. This regulatory mechanism reduces the overactivation of pro-inflammatory responses, thereby maintaining immune homeostasis [36,37]. Additionally, probiotic bacteria may exert an anti-inflammatory effect by secreting interleukin-10 (IL-10) through regulatory T cells [38]. 

Intestinal microbes stimulate gut macrophages in gut-associated lymphoid tissue (GALT) to express toll-like receptors (TLRs) [39]. Metabolic disturbances, commonly observed in schizophrenia (SCZ) patients, such as obesity, can compromise gut integrity, facilitating the activation of TLRs [40]. Bacterial metabolites, such as lipopolysaccharides (LPSs) and peptidoglycans (PGNs), mediate the immune response through TLRs and transmit information to the enteric nervous system (ENS) [41]. ENS, including its submucosal and myenteric ganglia, not only regulates barrier permeability, but also modulates the release of mediators that influence healing processes, epithelial proliferation, and differentiation [42]. Moreover, increased intestinal barrier permeability and TLR stimulation can lead to their overexpression, contributing to low-grade systemic inflammation in the host’s body [43]. Some authors suggest that gastrointestinal (GI) inflammation and immune responses play a crucial role in the pathogenesis of schizophrenia (SCZ). Abnormal immune system reactions may arise from microbial infections, food allergies, or disruptions in cell barriers [44]. This immune dysregulation potentially impacts the central nervous system (CNS), influencing behavior and cognitive function [43]. 

### 3.5. Autoimmune Diseases, Schizophrenia, and Gut–Brain Axis

The data report that the incidence of IBS in patients with schizophrenia can range from 17% in a study by Gupta et al. [45] to 19% (Garakani et al.) [46]. 

The diagnosis of Crohn’s disease uses the measurement of antibodies to Saccharomyces cerevisiae (ASCA). Studies conducted on patients with schizophrenia have indicated a significant association with elevated rates of this antibody (ASCA), particularly in those with a recent onset of the disease [47]. What is more, comparing the results of patients who had not previously taken antipsychotics with those taking medications, their ASCA levels were found to be significantly elevated [48].

Toxoplasma gondii infection during fetal or neonatal life is a known risk factor for early-onset schizophrenia [49]. Therefore, it is widely used in experimental models to generate gastrointestinal inflammation, causing intestinal dysbiosis and a state of increased gastrointestinal permeability [50]. Severance et al. (2015), in a study of clinical samples, found a correlation between *T. gondii* IgG levels and gastrointestinal IgG antigen in people with recently diagnosed schizophrenia [49]. There are reports according to which a history of celiac disease is a risk factor for developing schizophrenia in patients [51]. Antibodies to gliadin and transglutaminase (tTGA) are considered markers of celiac disease and gluten sensitivity. Eaton et al. detected moderate to high levels of these antibodies in patients with schizophrenia. These data suggest that intestinal inflammation caused by dietary antigens, bacterial infection, or dysbiosis is a close component in the course of schizophrenia [42]. 

### 3.6. Dysbiosis

The gastrointestinal tract is colonized with dynamically changing microbiota, predominantly composed of *Bacteroidetes* and *Firmicutes phyla*, with *Actinobacteria*, *Fusobacteria, Proteobacteria*, and *Verrucomicrobia phyla* making up the remainder [52]. 

Several factors can influence the composition of the microbiota, consequently affecting the gut–brain axis. These factors include stress, dietary habits, prior infections, and medication use (including antibiotics) [53]. The composition of gut microbiota depends on pre-birth conditions, mode of delivery, and changes over time with environmental factors [54]. Adopting a healthy diet, rich in plants and vegetables, can help reduce the abundance of pro-inflammatory bacteria, such as *Escherichia coli* and *Clostridium innocuum*, while promoting the growth of beneficial anaerobic bacteria, like *Faecalibacterium prausnitzii* and *Agathobaculum butyriciproducens* [55]. Furthermore, a diet high in fats can disrupt the delicate balance of the gut microbiota, leading to the inhibition of butyrate metabolism and potentially contributing to gut dysbiosis [56]. 

The beneficial role of the commensal gut bacteria relies on the maturation of the immune system, vitamin synthesis (i.e., vitamin K), and plays a protective role regarding pathogenic bacteria [57]. An imbalance in microbiome composition might influence CNS function and lead to a deterioration in mental well-being [7]. Studies on *Lactobacillus* and *Bifidobacterium* indicate their anti-inflammatory properties, reduction in cytokine levels, production of neurotropic factors, and improved antioxidant capacity [58]. 

Zheng et al. examined 63 patients with SCZ and 69 healthy controls, and found that the gut microbial composition between these two groups differed significantly. In medicated and unmedicated patients with SCZ, α-diversity (the number or distribution) was decreased. *Veillonellaceae* and *Lachnospiraceae taxa* also correlated with the severity of SCZ symptoms measured with The Positive and Negative Syndrome Scale Score (PANSS). Furthermore, a microbial panel consisting of *Aerococcaceae*, *Bifidobacteriaceae*, *Brucellaceae*, *Pasteurellaceae*, and *Rikenellaceae* presented a high probability of distinguishing SCZ-diagnosed patients and healthy controls. The next step in this study was fecal microbiota transplantation (FMT) from SCZ-diagnosed patients to germ-free mice. Scientists discovered that mice receiving microbiota transplants from SCZ-diagnosed patients demonstrated lower glutamate and higher glutamine and GABA concentrations in the hippocampus than the healthy controls. Animals presented disruptions in microbial genes and host metabolites, involved in the metabolism of amino acids and lipids, including glutamate, which is closely associated with schizophrenia pathogenesis. Moreover, these individuals were characterized by SCZ-like behavior. The authors of this study indicate the potential influence of microbiota on neurochemistry and neurologic function in the pathogenesis of SCZ [59]. 

There is also a growing interest in oropharyngeal microbiota, suggesting the potential existence of the oral–brain microbiota concept. Confirming the existence of a direct link between oral and gut microbiome and the brain is challenging and not fully established [60]. However, the study conducted by Zhu et al. involved the transfer of Streptococcus vestibularis, an oral bacterium, obtained from SCZ-diagnosed patients into the gut of mice with an antibiotic-induced reduction in gut microbiota. The researchers observed deficits in social behavior and alterations in neurotransmitter levels in the peripheral tissues, resembling those observed in patients diagnosed with schizophrenia [61]. 

In a study conducted by Ghaderi et al., 60 patients received vitamin D with a probiotic composition (*Lactobacillus reuteri*, *L. fermentum*, *L. acidophilus*, and *Bifidobacterium bifidum*). After 12 weeks of supplementation, the researchers observed notable improvements in PANSS scores and metabolic profiles [62]. 

Dickerson et al. reported no significant differences in PANSSs following 14 weeks of supplementation with *Lactobacillus rhamnosus* strain GG and *Bifidobacterium animalis* subsp. lactis strain Bb12 among 65 individuals diagnosed with SCZ [63]. Tomasik et al. conducted a placebo-controlled study administering *L. rhamnosus* GG and *B. animalis* subsp. lactis strain Bb12 for 14 weeks, but found no change in PANSSs after supplementation. These findings suggest that probiotic supplementation may not consistently impact symptom severity in individuals with schizophrenia [64]. 

Recently, there has been an increase in clinical and experimental studies debating the significance of the relationship between increased blood–brain barrier (BBB) permeability and schizophrenia. Critical to the proper functioning of the BBB is the integrity of neurovascular endothelial cells, which provide the homeostasis of the brain environment. BBB regulates the removal of toxins, nutrient delivery, and ionic balance. In addition, the BBB prevents the influx of inflammatory mediators and neuroactive substances [65]. K. Bechtera et al. (2010) [66] conducted a study on 63 patients with affective disorders and schizophrenia spectrum disorders and a control group of 4100 patients in whom albumin, IgG, IgA, IgM, oligoclonal IgG, and specific antibodies were analyzed in paired samples of cerebrospinal fluid (CSF) and serum. It was shown that 41% of psychiatric patients had CSF abnormalities that suggested increased BBB permeability. In addition, V. Braniste et al. (2014) [67] report that germ-free (GF) mice had lower expressions of the tight junction proteins occludin and claudin-5, which regulate barrier function in endothelial cells. The (GF) mice had increased BBB permeability compared to mice with normal intestinal flora.

Dysbiosis is further associated with impaired intestinal barrier integrity and increased permeability [68]. The penetration of intestinal pathogens into the bloodstream can activate the immune system and affect central nervous system (CNS) function. In addition, an increase in intestinal barrier permeability increases levels of pro-inflammatory cytokines, such as TNF-α and IL-1β, as well as IL-6, which further enhances the effects of TNF-α and IL-1β [42]. Smith et al. (2007) in their study used mice at 12.5 days of gestation that were injected with IL-6. The offspring of the mice exhibited abnormal behavior and showed an expression of genes resembling those found in schizophrenia patients [69,70]. Many studies report changes in bacterial taxonomic groups in patients with schizophrenia. Bacteria, such as *Fusobacterium*, *Lactobacillus*, *Megasphaera*, and *Prevotella*, most of which are Gram-negative, showed the greatest increase in numbers. Although Gram-negative bacteria are part of the normal microbiota, the increased permeability of the intestinal wall can cause the systemic circulation of intestinal inflammatory molecules, such as lipopolysaccharides (LPSs) [71]: Basta-Kaim et al. (2012) in their study administered LPSs to pregnant rat mothers subcutaneously at a dose of 1 mg/kg every other day from day 7 of gestation until delivery. Prenatally, LPS-treated rats showed changes in sensory gating and increased levels of pro-inflammatory cytokines (IL-1β, IL-2, IL-6, and TNF-α). In this and other studies, the presence of elevated LPSs in the bloodstream has been demonstrated as an effective neurodevelopmental model for SCZ in rodents [70]. 

Further research into interventions, such as prebiotics and probiotics, could improve outcomes for people with schizophrenia by restoring gut balance, reducing medication side effects, and improving overall health, given the potential impact of medication on the gut microbiota. However, to confirm a link between gut dysbiosis and schizophrenia, more studies are needed [9].

### 3.7. SCFAs and BDNF

In the process of enteric fermentation, bacteria produce short-chain fatty acids (SCFAs), i.e., butyrate, propionate, acetate, and valerate [72]. SCFAs are recognized as primary molecules responsible for the communication of the gut microbiome with the host [73]. SCFAs can cross the blood–brain barrier, where they stimulate cytokine production, modulate microglial activity, and exert an influence on CNS function [74]. Some research suggests that alterations in gut microbiota composition and SCFA production could influence brain function and behavior through the gut–brain axis [75]. 

Peng et al. indicated decreased concentrations of various SCFAs in SCZ-diagnosed patients and individuals at ultra-high risk compared to healthy controls (HCs) [76]. On the other hand, Li et al. did not observe significant differences in SCFA levels between patients with SCZ and HCs. However, scientists reported that treatment with risperidone for 24 weeks led to increased serum levels of butyric acid. This increase was correlated with reductions in positive symptoms and PANSSs in the drug-naive, first-episode patients [77]. 

Moreover, butyrate can promote the production of brain-derived neurotrophic factor (BDNF), which enhances neurotransmitter production through the vagus nerve [75]. BDNF is a neurotrophic factor crucial for neurodevelopment, neuroprotection, and the plasticity of glutamatergic and GABAergic synapses. It plays a significant role in neuronal differentiation and exerts an influence on serotonergic and dopaminergic neurotransmission [78]. Studies have indicated that individuals diagnosed with schizophrenia exhibit reduced blood levels of BDNF compared to healthy controls [79]. 

There is an apparent relationship between gut microflora and behavioral phenotype. Traditionally used probiotics include bacteria, such as *Lactobacilli* and *Bifidobacteria*, in their composition. A groundbreaking 2018 study by Berkley Luk et al. [80] showed that the colonization of gnotobiotic mice with a complex gut microflora that included four species of infant-type Bifidobacterium prevented behavioral disorders in mice. Adult mice lacking the microbiome were observed to have memory problems, anxiety-like behavior, and impaired motor skills. These disorders were not present in the behavior of mice colonized with *Bifidobacterium* spp., which additionally showed improved memory in colonized individuals. The study also shows that the administration of Bifidobacterium effectively modifies intestinal microflora and increases the production of short-chain fatty acids (SCFAs). SCFAs, such as butyric acid, can in turn act as a secondary messenger and prevent cognitive dysfunction [75]. Species such as *Roseburia* and *Faecalibacterium*, whose reduced levels have been found in people with schizophrenia, are responsible for butyrate production [81]. 

Dysbiosis in gut microbiota among SCZ-diagnosed patients could potentially affect SCFA production and, in turn, influence brain function [75]. However, the exact mechanisms linking SCFAs to schizophrenia pathogenesis remain unclear and require further investigation (Figure 1).

### 3.8. Microglial Cells

Disturbances in the microglial cells are considered one of the reasons for neurodevelopmental and neurodegenerative disorders. The role of gut microbiota is emphasized in microglial maturation and function [82]. In mouse models, the absence of gut bacteria and SCFAs led to disruptions in the development of microglial cells. Moreover, when mice were colonized with a diverse array of microorganisms, it resulted in enhanced functionality of microglial cells [7]. Since SCFAs act as substrates for energy production within mitochondria, alterations in SCFA levels can disturb oxidative processes and enhance sensitivity to oxidative stress [83,84]. 

In the context of schizophrenia development, significant attention is given to the impact of stress, particularly in susceptible individuals. Stress is recognized as a risk factor for various neuropsychiatric disorders, including schizophrenia [85]. Microglia cells possess the capability to transition between their pro-inflammatory and anti-inflammatory phenotypes in reaction to alterations in their surroundings, such as exposure to stress stimuli [86]. These cells influence the shaping of synaptic structures and function, particularly in modulating glutamatergic and GABAergic neurotransmission [87]. Studies have reported the presence of activated microglia in individuals diagnosed with schizophrenia, highlighting the potential role of stress-induced neuroinflammation in the pathogenesis of the disorder.

### 3.9. Kynurenic Acid (KYNA)

In the pathogenesis of schizophrenia (SCZ), the activation of the brain immune system manifests through elevated concentrations of kynurenic acid (KYNA) and proinflammatory cytokines in the brain and cerebrospinal fluid (CSF). KYNA acts as an antagonist of N-methyl-D-aspartate (NMDA) and α7 nicotinic acetylcholine receptors and is derived from astrocytes in the tryptophan degradation pathway [88]. Studies conducted on patients diagnosed with schizophrenia have revealed an elevated kynurenine/tryptophan ratio in the prefrontal cortex, indicating disruptions in serotoninergic transmission [89]. Furthermore, while the exact mechanism by which KYNA influences the central nervous system (CNS) remains unclear, it is hypothesized to modulate dopaminergic and glutaminergic transmission [90] (Figure 2). 

**Scheme 2 nutrients-16-02228-sch002:**
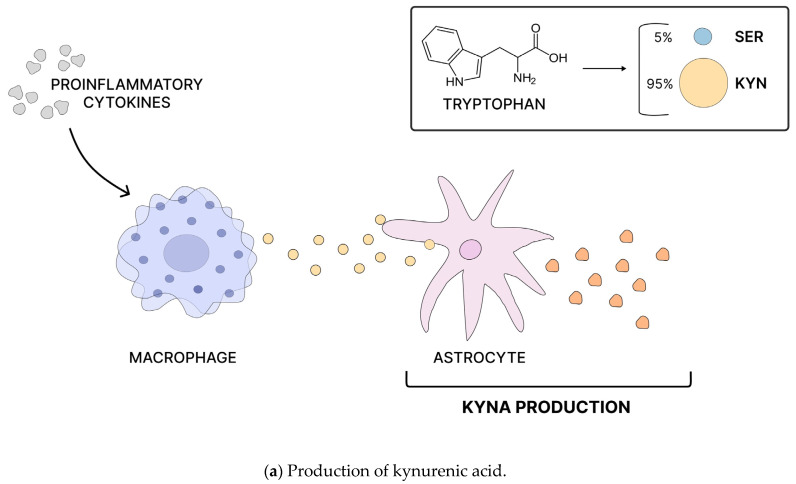
Production of kynurenic acid, increase in KYNA levels, and the pathogenesis of schizophrenia. Tryptophan is converted into serotonin (SER-in ~5%) and kynurenine (KYN ~95%). Kynurenine can then be metabolized into kynurenic acid (KYNA) and other metabolites. Proinflammatory cytokines stimulate circulating macrophages to produce more kynurenine, which astrocytes can later convert into KYNA (**a**). As an N-methyl-D-aspartate (NMDA) receptor antagonist, increased levels of KYNA can lead to NMDA receptor hypofunction in GABA interneurons, resulting in increased the glutamatergic activity of pyramidal neurons. This hyperactivity can over-activate the mesolimbic dopamine pathway, causing excessive dopamine release into the ventral striatum and potentially leading to psychosis (**b**) [91].

Zhu et al. conducted a study in which male mice were transplanted with fecal microbiota from drug-free patients diagnosed with schizophrenia. The animals exhibited altered behavior, including psychomotor hyperactivity, and showed impairments in learning and memory. Furthermore, an increase in the kynurenine–kynurenic acid pathway of tryptophan degradation was observed peripherally and in these mice’s brains. Additionally, basal extracellular dopamine levels in the prefrontal cortex and serotonin (5-HT) levels in the hippocampus were found to be elevated compared to mice receiving transplants from healthy controls [92]. Despite understanding the gaps in KYNA’s role in the pathogenesis of schizophrenia, this area holds promise for further research and potential therapeutic interventions.

In Table 1, we summarize the importance of the axis in mental disorders.

## 4. Diets

A study of a group of more than 1000 college students underscores the importance of diet in the context of mental health. Educational interventions and the continuous updating of knowledge on the relationship between diet and mental health are essential, as the study showed that unhealthy diets are associated with a higher prevalence of anxiety, depression, and stress symptoms, and that abnormal dietary patterns are significantly associated with neuropsychiatric abnormalities. Excessive intake of carbohydrate-rich foods and low intake of dairy products have been associated with an increased incidence of psychiatric disorders (depression and anxiety) and sleeping problems. Modern industrialization has increased the risk of obesity and related diseases through the production of high-calorie foods. The mass production and widespread availability of such products promote overweight and obesity. This can have a detrimental effect on mental health, including the risk of schizophrenia [93]. 

### 4.1. Mediterranean Diet 

A promising trend in treating neurodegenerative and neuropsychiatric disorders is the anti-inflammatory diet, which may also benefit schizophrenia patients. The Mediterranean diet, rich in vegetable oils and low in saturated fats, exemplifies this approach. Originating in Greece and southern Italy, it aims to improve lipid profiles, normalize blood pressure, and reduce cardiovascular risk [94]. 

A study was conducted on the dietary habits of three groups of patients: those with the schizophrenia-deficit subtype (SCZ-D), those with the non-deficit subtype (SCZ-ND), and healthy subjects (HCs). According to the authors of the study, patients with schizophrenia deficit (SCZ-D) did not adhere to the Mediterranean diet, mainly due to several factors. The first may be the presence of negative symptoms of schizophrenia, such as apathy, decreased energy, and difficulty concentrating. In addition, the impact of a schizophrenia diagnosis on patients’ eating habits and a lack of awareness of the benefits of healthy eating are important factors. According to the authors Kowalski et al., there is some evidence to suggest that better adherence to the Mediterranean diet may benefit cognitive function in a population unrelated to psychiatric problems. However, to date, there have been no studies that have focused on testing this hypothesis in the context of people with schizophrenia [95]. Costa et al. also noted that the eating habits of schizophrenics, such as low fiber and folate intake, and active smoking were found to negatively affect diet quality in addition to the effects of the diet itself. This underscores the complexity of treating schizophrenia, where the effects of diet, physical activity, and patient education on weight control are all crucial components [96]. 

Therefore, although reports on the beneficial effects of the Mediterranean diet on the course of schizophrenia are inconclusive, its effectiveness may be enhanced by using other interventions in parallel (Table 2) [97]. 

### 4.2. DASH Diet

The DASH (Dietary Approaches to Stop Hypertension) diet is a growing dietary trend. It is an eating plan that emphasizes the consumption of large amounts of fruits, vegetables, low-fat dairy products, fish, poultry, nuts, and grains. It also limits the intake of sweets, sugary drinks, red meat, saturated fat, and cholesterol. The beneficial effects of the DASH diet were discovered more than two decades ago. This discovery had a significant impact on the approach to blood pressure control, as significant reductions in both systolic and diastolic blood pressure were noted [98]. 

A systematic review looked at 16 studies on the effects of the DASH diet on patients’ mental health. It concluded that the DASH diet had a positive effect on schizophrenia patients. Different methods used to analyze the data and different mental states of the patients led to conflicting results: five trials found a negative association between adherence to the DASH diet and psychiatric disorders, while four trials found a positive effect. Overall, the DASH diet is likely to be beneficial for mental well-being, but the inconclusive results may be due to methodological differences and the mental state of the patients [99]. 

In patients diagnosed with schizophrenia and metabolic syndrome (MetS), it has been recognized in recent years that the DASH diet may have a beneficial effect. In a three-month study of patients with schizophrenia and metabolic syndrome, the effects of the DASH calorie-restricted diet were compared with those of a standard hospital diet. The results suggest that the DASH diet does not have a significant effect on metabolic syndrome. However, it did improve diet quality in hospitalized schizophrenic patients with MetS.

### 4.3. Ketogenic Diet

The ketogenic diet is used not only for weight loss, but also for the improvement of the patient’s lipid profiles, as the body is put into a state of ketosis. This state leads to a decrease in glucose levels, resulting in the production of ketone bodies (Figure 3). The presence of ketone bodies is an exponent of the effectiveness of the diet. This diet plan includes 1 g of protein per kilogram of body weight, 10–15 g of carbohydrates per day, with the remaining calories coming from fats [100]. This diet increases the risk of heart disease and atherosclerosis by decreasing glycated hemoglobin and increasing low-density lipoprotein (LDL) [101]. 

The ketogenic diet reverses impaired glucose metabolism, inflammatory response, oxidative stress, and the dysregulation of neurotransmitter homeostasis. Some studies also suggest improvements in monoaminergic systems and the hypothalamic–pituitary–adrenal axis. These are systems that are often implicated in psychiatric disorders [102]. 

The mechanism by which glucose metabolism may have an impact on brain function in people with schizophrenia has long been the subject of research interest. One theory to explain this mechanism is that of bioenergetic dysfunction. According to this hypothesis, abnormalities in glucose metabolism lead to a disrupted energy supply to brain cells, resulting in the dysfunction of the nervous system. The brain is an organ that is highly susceptible to energy deprivation, consuming about 20% of the body’s energy while making up only 2% of our body weight [103]. Abnormalities in glucose metabolism may therefore lead to a reduction in energy use by brain cells, which may have an impact on the changes in cognitive function observed in people with schizophrenia. The existence of a link between glucose metabolism and brain function in people with schizophrenia is an area of research that needs further observation to fully understand the pathophysiological mechanisms of this disease [104].

Adult patients with severe, treatment-resistant psychiatric disorders (major depressive disorder, bipolar disorder, and schizoaffective disorder) were studied in a retrospective clinical trial. Patients with schizophrenia have been on a restrictive ketogenic diet during their stay in a psychiatric hospital. The measure of improvement was the Positive and Negative Syndrome Scale (PANSS). Significant improvements in metabolic parameters were observed. These included weight loss, normalization of blood pressure, and regulation of blood glucose and triglyceride levels [105]. The study therefore suggests that a restrictive ketogenic diet may be an effective therapy for patients with refractory psychiatric disorders, including schizophrenia. It is associated with improvements in metabolic parameters and symptom scores on the PANSS.

Dietcha et al. found a potentially beneficial effect of a ketogenic diet on anxiety disorders in patients diagnosed with schizophrenia. It should be noted, however, that this was not high-quality evidence [106].

In conclusion, current evidence on the efficacy of the ketogenic diet for the treatment of psychiatric disorders is inconclusive. It is still unclear whether a ketogenic diet on its own or in combination with a low-carbohydrate diet is the best way to achieve therapeutic effects. In addition, a thorough characterization of the adverse effects and the potential for recurrence of symptoms after the discontinuation of the diet is crucial. 

### 4.4. Dietary Caffeine

The alkaloid chemical compound caffeine (1,3,7-trimethylxanthine) exerts its primary action through the antagonism of adenosine A1, A2A, A3, and A2B receptors. Caffeine’s mechanism of action involves competitive binding to these receptors, blocking their activation by the endogenous ligand, adenosine, which can affect glutamatergic and dopaminergic transmission [107,108,109]. Adenosine receptor imbalance can increase glutamate sensitivity and a loss of control over dopamine activity, which is clinically significant. Adenosine receptors are part of an integral neuromodulatory system in the brain that influences neurocognition, remembering, and learning. A2A receptors attenuate the effects of D2 receptor-dependent molecules, suggesting that A2A receptor blockade may be useful for the treatment of schizophrenia [107]. Dipyridamole, an adenosine transporter inhibitor, is an example of a schizophrenia drug that increases adenosine levels in extracellular space [110].

The overconsumption of caffeine is a common occurrence in the psychiatric population, especially in the context of schizophrenia [111]. Typical characteristics of caffeine dependence, such as the tolerance to this substance and withdrawal symptoms, are closely related to the occurrence of depressive episodes, antisocial disorders in adults, and generalized anxiety disorder [112]. 

There is also a belief that excessive caffeine consumption may worsen chronic mental disorders [113]. The high dose of caffeine used in patients with schizophrenia may affect the severity of dyskinesia symptoms through its antagonistic effect on A2 receptors by the upregulation of dopamine receptors. In addition, caffeine withdrawal may help alleviate dyskinesia symptoms [114]. Low-to-moderate doses of caffeine have been documented as a cognitive enhancer in patients with schizophrenia [115]. In contrast, another study showed that caffeine administered at relatively high doses, more than 250 mg per day, may have an antipsychotic effect [116]. 

Administration of a 10 mg/kg dose of intravenous caffeine improved mood and reduced withdrawal symptoms without increasing anxiety, according to a study investigating the effects of caffeine on schizophrenia [117]. However, the study had important limitations: the lack of a control group and the use of very high doses of caffeine, which can induce “positive symptoms” in people who do not have psychosis.

Caffeine may interact with other drugs, such as affecting the blood levels of clozapine relative to the dose administered [118].

A clinical case has been reported of a patient with chronic clozapine-treated schizophrenia who was a heavy consumer of caffeinated energy drinks (four cans of Red Bull per day). The patient’s condition suddenly deteriorated. He was admitted to the emergency department with symptoms of acute respiratory failure and altered mental status. Diagnostic studies revealed severe metabolic acidosis, elevated inflammatory parameters, and acute kidney injury that was attributed to inflammation of the kidneys. There are several reasons why the interaction between caffeine and clozapine may lead to kidney damage. Caffeine inhibits an enzyme that metabolizes clozapine, which can increase its blood levels and potentially cause toxicity. In addition, a high caffeine intake can lead to dehydration, which increases the risk of kidney stones, and a reduction in blood flow to the kidneys. Ultimately, both changes in the metabolism of clozapine and its effects on the body may result in an interaction between these substances. It was concluded that severe toxicity and multiple organ failure during clozapine treatment may have been caused by a high caffeine intake of more than 600 mg per day [119]. 

Therefore, there is clinical evidence that caffeine may be beneficial to patients with schizophrenia, especially for cognitive impairment and negative symptoms, through its effect on the A2 receptor. However, for patients with positive symptoms, it is important to control the dose of caffeine. It is also worth noting that caffeine can have a variety of effects depending on the individual characteristics of the patient, such as gender or the presence of other coexisting diseases [120], so it is recommended to be cautious about the consumption of caffeine by people diagnosed with schizophrenia.

### 4.5. Gluten and Schizophrenia

Celiac disease is twice as common in patients with schizophrenia as in the general population, according to epidemiological studies [121]. 

The elimination of gluten from the diet may be beneficial for patients with a diagnosis of schizophrenia [122]. As early as World War II, observations by Joseph Dohan revealed an important association between average annual wheat consumption and an increase in first hospitalizations for schizophrenia in women [123]. In addition, the same psychiatrist noted that a lower incidence of schizophrenia was observed in places where wheat consumption was considered limited, compared to areas of the world where gluten-rich products were more accessible [124]. 

A case has been described in which single-photon computed tomography (SPECT) showed prefrontal cortex dysfunction in a patient with celiac disease and a diagnosis of schizophrenia, which disappeared after the introduction of a gluten-free diet. Hypoperfusion of the left frontal lobe without visible structural changes was observed on a SPECT scan. Atrophy and anti-endomysial antibodies were found on an intestinal biopsy. The resolution of psychiatric symptoms and standardization of histologic and SPECT findings were achieved by initiating a gluten-free diet without adjunctive psychiatric pharmacotherapy. Previous neuroleptic pharmacotherapy was discontinued. There was no recurrence of schizophrenia symptoms in the patient on the gluten-free diet after the discontinuation of pharmacological treatment. Therefore, it can be concluded that the gluten-free diet was the only therapy that led to improvements in his psychiatric and physical condition [125]. 

The efficacy of the gluten-free diet (GFD) was studied in two schizophrenia-positive patients with tissue transglutaminase (tTG) or antiglandin (AGA) antibodies. Both patients experienced improvements in psychiatric symptoms. These included anxiety, hallucinations, delusions, and disorganized thinking. Negative and extrapyramidal symptoms also decreased. The results suggest that a gluten-free diet may be beneficial for people who have been diagnosed with schizophrenia, particularly those who have antibodies to tTG and AGA [126]. 

Genois et al. describe the case of a 23-year-old woman with long-standing visual and auditory hallucinations that improved completely with the introduction of a gluten-free diet. However, symptoms returned after the accidental ingestion of gluten. TTG antibody tests were negative. Therefore, it is not clear whether the patient had a visceral disease or a hypersensitivity to gluten. Nevertheless, the elimination of gluten proved to be effective and safe in alleviating his psychotic symptoms. This confirms the beneficial effect of the gluten-free diet on schizophrenia patients [127]. 

A systematic review by Aranburu et al. found that gluten avoidance may improve cognitive function in schizophrenia [128]. The study included 16 schizophrenic or schizoaffective patients without celiac disease, but with high levels of antigliadin IgG. The participants who were on a gluten-free diet were divided into two groups: one group was provided additional milk cocktails with gluten flour, and the other group with rice flour. There was no improvement in positive symptoms in patients receiving the rice flour cocktail. However, an improvement in negative symptoms was observed. SANS (Scale for the Assessment of Negative Symptoms) scores showed improvements in five domains: affective impairment, alogy, abulia/apathy, anhedonia, frustration, and attention deficit [128]. 

Gluten consumption can cause inflammatory reactions in the brain. This increases the permeability of the blood–brain barrier. In patients with schizophrenia, this can increase the penetration of haptoglobin-2 and worsen symptoms. Due to the increased permeability of the blood–brain barrier caused by the inflammatory process, the concentration of this peptide is increased in schizophrenia patients [129]. Studies have shown that the effect of a gluten-free diet on schizophrenia symptoms is either positive or neutral. Based on these data, it can be concluded that a gluten-free diet is well tolerated and can be used to treat schizophrenia. However, since the diagnosis and treatment of celiac disease in the presence of coexisting conditions appear to significantly improve patient prognosis, attention should be paid to gastric symptoms suggestive of celiac disease in clinical practice.

### 4.6. Diet of Diabetic Patients

Diabetes is a serious concern for people with schizophrenia, as they are at an increased risk of developing the disease when compared to the general population. The basic information about glucose metabolism is presented in Figure 4. A study analyzed data from both the World Health Organization and the Food and Agriculture Organization (FAO). It was found that a more severe course of schizophrenia, as manifested by the overall poor health of the patients, was associated with the consumption of products rich in saturated fatty acids relative to those rich in polyunsaturated fatty acids. This means that a higher percentage of fat, lower indicators of diet quality, and a higher intake of unhealthy products, such as full-fat creams and sweetened beverages, were found in schizophrenia patients who preferred such a diet. There is an association between these poor dietary habits and poorer health outcomes in patients with schizophrenia. However, these outcomes may also be influenced by other factors, such as medications and overall health [130]. 

In light of the epidemiologic studies conducted by Stokes et al., an important association between sugar intake and the severity of schizophrenia symptoms was confirmed, particularly in the context of drug treatment, with a focus on the use of clozapine. In addition, a secondary analysis suggested a possible association between the intake of polyunsaturated fatty acids and the worsening of the symptoms of schizophrenia, independent of the pharmacological treatment [131]. 

Insulin resistance has been diagnosed in patients with schizophrenia and type II diabetes, both in those on medication and in untreated patients [132]. Furthermore, the importance of the risk of developing type II diabetes is underscored by the meta-analysis by Philinger et al. who suggested that glucose homeostasis is impaired from the first psychotic episode in people with schizophrenia [133]. In addition, the study by Huang et al. showed that the incidence of type II diabetes in schizophrenics is significantly higher than in the general population, especially in people under the age of 60 years. This may indicate that it is not related to a decrease in metabolic rate with age [134].

Furthermore, Upadhyaya et al. suggest structural abnormalities in the first psychotic episode [135], and Leticia-González-Blanco et al. show prolactin dysregulation in schizophrenia [136]. This hormone can therefore be used as a hormone marker for the prognosis of schizophrenia [137]. 

Some components of the diet, i.e., simple sugars and saturated/trans fats, can impair glucose and insulin metabolism, which indirectly affects pituitary function, and may lead to increased cortisol secretion. Glucocorticoids increase glucose production in the liver and decrease glucose uptake in other tissues. Prolonged use of glucocorticoids leads to hyperglycemia and insulin resistance [138]. 

Eating more artificial foods was found to have a significant effect on the expression of genes associated with pituitary function, including those responsible for glycemic control and glucose metabolism, in studies of the effects of diet on pituitary and neuroendocrine regulation in patients with schizophrenia [75]. A diet high in saturated fat and carbohydrates may be detrimental to the prognosis of these patients. It may increase the risks of insulin resistance and metabolic syndrome. In contrast, glycemic control and pituitary function may be improved with a diet low in saturated fat, high in fiber, and low-glycemic-index products. Because glucose homeostasis is impaired in schizophrenia patients from the first psychotic episode, the study emphasizes the need to monitor glycemic levels in schizophrenia patients and recommends further research on the effects of diet on pituitary neuroendocrine regulation in these patients.

## 5. Supplementation

Traditional medications employed for schizophrenia may not always deliver satisfactory results [139]. Due to this fact, supplementary treatments reinforcing the positive effects of conventional therapies are gaining more attention. The World Federation of Societies of Biological Psychiatry (WFSBP) and the Canadian Network for Mood and Anxiety Disorders (CANMAT) recently proposed new global guidelines emphasizing the use of nutraceuticals and phytoceuticals for treating psychiatric disorders. Among the 35 substances reviewed, five were evaluated for their potential as adjunctive therapies in the treatment of schizophrenia. N-acetylcysteine, folates, and Ginkgo biloba emerged as the most effective, earning recommendations for supplementary use for psychotic disorders. In contrast, omega-3 fatty acids and vitamin D were not recommended [140].

### 5.1. Omega-3 Fatty Acids

Omega-3 fatty acids play a crucial role in supporting proper neurodevelopment and brain function [141]. Eicosapentaenoic acid (EPA) and docosahexaenoic acid (DHA) can modulate serotonin production in the brain. This has a direct impact on the regulation of executive functions, social behavior, and sensory receptor blockade, and consequently on the dysfunctions observed in schizophrenia patients. DHA increases the fluidity of the cell membrane in postsynaptic neurons. EPA works by reducing the release of prostaglandin E2 (PGE2), which in turn leads to an increase in serotonin release [142]. Since a decrease in PGE1 activity may be associated with an increase in dopamine release, a deficiency of PGE1 is consistent with the dopamine hypothesis of schizophrenia [143]. 

Polyunsaturated fatty acids (PUFAs) and their metabolites play an important role in the suppression of the inflammatory response by reducing the production of cytokines and by exerting neuroprotective functions [144]. The excessive secretion of pro-inflammatory cytokines may result from disorders in the synthesis of polyunsaturated fatty acids. The lack of controlled regulation by PUFAs and their anti-inflammatory metabolites may predispose patients to neuronal damage [145]. Some researchers propose that individuals with schizophrenia (SCZ) may experience disturbances in lipid homeostasis, potentially leading to a decrease in polyunsaturated fatty acid (PUFA) levels [146]. 

Tang et al. [147] evaluated the impact of omega-3 supplementation in SZ patients (720 mg/day). They observed improvements in negative, general, and total PANSS subscores, but they were not statistically significant. They, however, detected a meaningful improvement in the delayed memory factor in the Repeatable Battery for the Assessment of Neuropsychological Status (RBANS) assessment. Jamilian et al. [148] also assessed the effects of the omega-3 supplementation (1 g/day). They found that it notably decreased total and general PANSSs, but not positive and negative scores. Those findings were later confirmed by Pawełczyk et al. [149], whose research focused on EPA + DHA (2.2 g/day) intervention in SZ patients. To counter the above, Qiao et al. [150] claim that omega-3 supplementation along with traditional antipsychotic therapy provides results no better than in the placebo group considering the PANSS scale. It is important to note that their study employed relatively low doses of EPA + DHA (1 g/day). This factor could potentially have impacted the observed outcomes.

In summary, most studies suggest that omega-3 supplementation, along with traditional therapy, has mild to moderate positive effects. The results are statistically more significant when higher doses of acids are used. However, it is crucial to note that further research is required to establish the optimal dosage and usage conditions. Additionally, confirming these findings with larger sample sizes would contribute to a more comprehensive understanding of the potential benefits.

### 5.2. N-Acetylcysteine (NAC)

Glutathione (GSH) dysregulation is reported to be one of the factors linked to raised oxidative stress in SZ. N-acetylcysteine (NAC) is an antioxidant drug and a precursor of L-cysteine, which donates cysteine for GSH synthesis. Studies confirm lowered levels of GSH in SZ patients, which is also linked to NMDAR hypoactivity. This is further connected to increased ROS production and oxidative stress [151]. This is why NAC supplementation is being studied for symptom improvement in the context of SZ. However, existing research is unclear regarding the impact on PANSS results after NAC intervention in SZ patients.

Numerous studies support the positive impact of N-acetylcysteine (NAC) supplementation on schizophrenia symptoms [152,153,154]. Sepehrmanesh et al. [155] observed improvements in positive, negative, general, and total subscales of PANSSs, along with enhancements in certain cognitive functions.

M. Rapado-Castro et al. [156] confirmed the positive impact on the negative scale, but also provided an interesting conclusion. Participants with >20 years of illness duration present much better improvements on all scales than ones with a shorter illness duration.

While the above studies confirm improvements in the negative subscale, consistent improvements are not found in other scales.

Conversely, several studies fail to conclusively demonstrate the beneficial effects of N-acetylcysteine (NAC) supplementation in SZ.

E. Neill et al. [157], focusing on clozapine-resistant patients, did not observe any enhancements in the negative PANSS scale, cognitive functions, or overall quality of life. Additionally, Conus et al. [158] found no PANSS improvements in the general SZ population. Nevertheless, intriguingly, individuals with a specific level of redox dysregulation (GPxBC activity > 22.3 U/g of Hb) did experience improvements in processing speed and positive symptoms.

The analyzed research does not allow us to draw definitive conclusions. However, the mentioned studies suggest that there may be a subtype of patients who respond particularly well to NAC, especially when the disease has not yet led to significant organic changes in the brain, as in the study by Neil et al. [157]. While NAC supplementation appears to offer potential benefits, it seems that several conditions need to be met for it to be effective. Further research on larger samples is required to determine additional concrete considerations when administering NAC.

### 5.3. Withania somnifera Extract (WSE)

*Withania somnifera*, a herb widely known as Ashwagandha, is characteristic of the Ayurvedic, Siddha, and Unani Traditional Systems of Medicine [159]. The most promising effects of WSE on the nervous system are connected with its immunomodulatory, anti-inflammatory, and neuroprotective activities [160,161,162], which are connected with an abundance of phytochemical compounds found in the plant [163]. A decrease in GABAergic signaling is one of the speculated contributors to schizophrenia [164]. Traditional WSE showed a GABAergic signaling enhancement potential, which was expressed by the affinity for the GABAA and GABAρ receptors [165].

Moreover, other stated factors taking place in the development of SZ are immuno- and neuroinflammation [166,167]. WSE reveals a potential for COX-2 and NF-κB inflammatory signaling pathway inhibition [168,169], which might be an effective supportive mechanism of action in the treatment of schizophrenia [170]. Celecoxib, a COX-2 inhibitor, has shown promising results when used as an additional treatment alongside risperidone in patients experiencing their first episode of schizophrenia [171]. Therefore, it may be beneficial to explore the potential effectiveness of WSE usage, given its minimal side effects [172].

Chengappa et al. [173] decided to examine if WSE might be a valuable adjunctive to traditional therapy for patients suffering from an exacerbation of schizophrenia symptoms. Outcomes showed that the WSE group was more likely to achieve improvements of 20% or more in negative, general, and total symptom PANSSs compared to the placebo group, although this did not apply to positive symptom scores. The first improvement was noticeable after 4 weeks of treatment. However, IL-6, hs-CRP, and S100B levels showed only non-significant differences between both groups. Further analyses conducted by Gannon et al. [174] proved the effectiveness of WSE in lessening the symptoms of depression and anxiety in exacerbated schizophrenia, although their study was designed mainly for patients with an exacerbation of positive symptoms rather than for symptoms of depression and/or anxiety. Therefore, they did not use scales specifically designed for schizophrenia depression assessment.

It is crucial to acknowledge that the existing body of research is limited and there are not enough comprehensive studies to fully assess the validity and long-term efficacy of WSE usage in managing schizophrenia. To solidly validate these results, a larger prospective study is necessary. Future research could assess cognitive benefits in a more stable patient group, extending study durations and incorporating functional assessments [173,174]. Additionally, investigations should determine optimal dosage levels to ensure the safety and efficacy of WSE as a potential treatment for schizophrenia, alongside symptoms such as anxiety and depression.

### 5.4. Vitamin D (VD)

Despite its role in proper neurodevelopment [175], vitamin D may also affect neurotrophism, neurotransmission, and neuroplasticity [176]. Recent studies showed that patients with SZ are more prone to decreased levels of VD [177]. Moreover, VD hypovitaminosis in neonates was correlated with a higher risk of developing SZ [178].

Sheikhmoonesi et al. [179] studied the impact of VD supplementation on 80 male SZ VD-deficient patients. Subjects received a single 600,000 IU injection at the beginning of the experiment, along with their regular antipsychotic treatment. They did not detect any change in PANSSs compared to the control group after 3 months.

Krivoy et al. [180] also did not confirm any significant improvement in the PANSSs of 44 severely ill (total PANSS > 70) patients after administering 14,000 IU per week, over 8 weeks.

On the other hand, Neriman et al. [181] used a more sophisticated dosing strategy on 40 patients. Deficient (VD < 20 ng/mL) patients received 50,000 IU once a week. Insufficient (VD 20–29.99 ng/mL) patients received 1500 IU daily (10,500 weekly total). Additionally, patients who were still presenting an insufficiency (<30 ng/mL) were administered additional doses. Supplementation lasted 8 weeks. Researchers detected significantly better results for SANS and SAPS after VD supplementation, but they did not evaluate the type of antipsychotics or other medication used during the trial.

There are very few studies focusing on VD intervention in SZ therapy. Additionally, they are all performed on small samples. Based on the database, it is possible to notice that proper dosing regimens have to be used to obtain the most beneficial effects, which might be more noticeable, only in less severe cases.

### 5.5. Vitamins E and C

There was no potential effect of vitamin E supplementation on extrapyramidal symptoms based on a meta-analysis conducted by Stubbs et al. [182]. However, positive antioxidant effects of vitamin C have been observed in schizophrenia patients. In addition, retrospective analyses have shown that antioxidants (vitamins C and E) are most effective in patients who are taking lower doses of antipsychotic drugs.

In patients with acute schizophrenic episodes who had low baseline levels of polyunsaturated fatty acids, separate administrations of omega-3 fatty acids, vitamin C, and vitamin E at moderately high doses were shown to induce psychotic symptoms. However, it was concluded that combining these appeared to be both safe and more effective [183].

### 5.6. Foliates

People with schizophrenia have significantly higher levels of homocysteine (Hcy) than the healthy population [184]. Hcy is found to induce oxidative stress and apoptosis. Moreover, it triggers vascular damage and mitochondrial dysfunction. Foliate is an essential molecule required in the methionine synthesis pathway, where Hcy is utilized. Inadequate levels of folate, as seen in patients with schizophrenia [185], could potentially inhibit this pathway, resulting in hyperhomocysteinemia [186,187].

A 2013 trial by Roffmann et al. [188] on 140 participants concluded that folate plus vitamin B12 supplementation improves the negative symptoms of SZ (SANS). This positive result was present only for patients carrying the homozygotic folate hydrolase 1(FOLH1) 484T allele. This is a high-functioning variant of FOLH1, a gene regulating the intestinal absorption of dietary folates. Patients with 484C, which is a low-functioning variant, did not show strong improvements.

Hill et al. [189] performed a trial on 28 participants to determine the impact of the methylenetetrahydrofolate reductase (MTHFR) 677C>T single nucleotide polymorphism (SNP) on folate supplementation results. The activity of MTHFR is crucial for proper folate availability as it activates folate by catalyzing the reduction of 5,10-methylenetetrahydrofolate to 5-methyltetrahydrofolate. Each 677T variant of this gene reduces MTHFR activity [190] and its expression might increase the risk of schizophrenia [191].

Although the overall improvement was non-significant compared with the placebo group (likely due to the small sample), patients with 677T alleles showed significantly better results than 677C homozygotes.

Another study by Roffmann et al. [192] was a controlled trial based on administering L-methyl folate (levomefolic acid). The experimental group experienced improvements in PANSSs, as well as changes in MRI-tested brain activity. As opposed to folic acid supplementation, this study observed positive results regardless of the patient’s genotype.

In contrast, K. Allott and colleagues [193] explored the impacts of supplementing vitamins B12, B6, and folic acid on early-stage schizophrenia (First Episode Psychosis—FEP). While the study did observe a decrease in homocysteine levels in the treatment group, it did not conclusively find any significant positive effects of the supplementation on schizophrenia symptoms, except for attention/vigilance, particularly in female patients. This finding remains consistent across different genotypes. One possible explanation for this could be that the baseline symptom levels in this study were significantly lower than those in the referenced studies.

Foliate supplementation in schizophrenia patients is still an under-researched topic, and potential therapeutic effects might be dependent on the patient’s genotype. Therefore, to maximize the benefits, it is advisable to consider genetic factors when selecting a specific substance.

### 5.7. Selenium

Patients suffering from schizophrenia have lower serum levels of selenium when compared to the healthy population [194,195]. This is reported to be associated with metabolic syndrome components in SZ patients [196].

Selenium plays a key role in human organisms; it is the crucial building block of selenoproteins. One of them is the glutathione peroxidase (GPx) antioxidant system, which converts hydroxyl radicals and peroxides into nontoxic compounds. Selenoproteins are also important for proper brain function. Selenoprotein P (SEPP1) takes part in delivering selenium to the brain and has neuroprotective characteristics—enhances survival and reduces the apoptotic death of neurons [197]. To add to this, the impaired function of GPx and raised oxidative stress are linked to SZ occurrence [198,199]. This suggests that supplementing Se might positively impact SZ patients.

According to the study mentioned above, conducted by Hamidreza Jamilian and Amir Ghaderi [200], the supplementation of selenium and probiotics showed a positive impact on the PANSSs and certain metabolic profiles of patients. However, a limitation of the study is its small participant number. Additionally, it remains challenging to discern the specific role played by either selenium or probiotics in producing the observed effects.

In a study by Li et al. [201], involving 158 subjects with schizophrenia (SZ), the researchers conducted an experiment involving a 3-month supplementation of ‘Selenium weikang’. The results revealed that the supplementation not only increased selenium levels along with other trace elements, but also surprisingly led to a decrease in arsenic levels. The primary focus of the study was on changes in trace element levels following the supplementation period. Notably, beyond these findings, it was observed that 60% of female and 18% of male patients reported experiencing increased appetite and improved memory after the treatment.

Even though decreased Se level in SZ patients is a confirmed phenomenon [195], research on its supplementation therapeutical effects remains scarce. The results of existing studies are promising, but to determine its effects on schizophrenia symptoms, more research is required.

### 5.8. Ginkgo Biloba Extract (GBE)

Ginkgo biloba (G. biloba) is a herb often linked to traditional Chinese medicine and recognized for its wide range of therapeutic applications. The potential efficacy is connected to unique compounds found in G. biloba leaves, such as terpenoids (ginkgolides and bilobalide), ginkgolic acid, flavonoids (kaempferol, quercetin, and isorhamnetin), and bioflavonoids (sciadopitysin, ginkgetin, and isoginkgetin) [202].

The majority of these substances contribute to the neuroprotective effects of GBE, which consist of anti-oxidative [203] and free radical scavenging properties [204]. Both of these mechanisms are suspected to have a potential impact on schizophrenia symptom reduction, due to the correlation between oxidative damage and schizophrenia severity [205,206]. The enhanced production of reactive oxygen species (ROS) might be the effect of defective dopamine metabolism [207]. Moreover, dysfunctional glutamatergic signaling can result in decreased glutathione production, which might be the cause of NMDA receptor hypofunction [208]. The outcomes of this process might be increased free radical production and oxidative damage [209].

X. Y. Zhang et al. [210] decided to evaluate the validity of GBE supplementation in addition to traditional haloperidol treatment and its impact on one of the scavenging enzyme levels—superoxide dismutase (SOD)—in patients with treatment-resistant schizophrenia. After 12 weeks of therapy, the GBE group showed significant decreases in SAPS and SANS scores, while only SANS scores decreased significantly in the non-GBE group. Both groups experienced reductions in Treatment Emergent Symptom Scale (TESS) scores, but improvements were more pronounced in the GBE group, particularly in subscores 1 and 3. Initially, elevated SOD levels decreased significantly only in the GBE group by the end of the study. The findings indicate that GBE supplementation might increase the therapeutic potential of haloperidol by reducing the damage caused by free radicals.

In a study by Doruk et al. [211], researchers examined the effects of atypical antipsychotic clozapine therapy with adjunctive GBE in the treatment of refractory schizophrenia. After an evaluation of the GBE group, only the SANS score showed a significant reduction. Both BPRS and SAPS scores did not show improvements when compared to the control group. The results lead to the conclusion that GBE supplementary therapy might reduce negative symptoms of the disorder. However, researchers emphasized that the dose of GBE used during the study (120 mg/day) could be insufficient.

W. F. Zhang et al. [212] evaluated the impact of complementary GBE supplementation on the management of Tardive Dyskinesia, a hyperkinetic movement disorder connected to the administration of psychotropic drugs used in schizophrenia therapy. It is characterized by repetitive and involuntary movements of the oral area or extremities [213]. After 12 weeks of treatment, the Abnormal Involuntary Movement Scale (AIMS) score significantly decreased in comparison to the placebo group, which indicates that GBE might be an effective adjunct that might reduce the side effects of conventional schizophrenia treatment.

In summary, Ginkgo biloba supplementation alongside traditional antipsychotic treatments shows potential in reducing positive and negative symptoms, as well as improving treatment-emergent symptoms. However, existing studies have limitations, notably their very small sample sizes. While these findings are encouraging, further research with larger and more diverse populations is necessary for conclusive results on the efficacy of Ginkgo biloba supplementation in schizophrenia treatment.

## 6. Impacts of Socioeconomic Factors, Nicotine, and Alcohol on Schizophrenia Patients 

### 6.1. Socioeconomic Status

Among schizophrenia patients, the rates of loneliness (patients 52%; control group 21%) and unemployment (patients 77%; control group 41%) are higher, both of which increase the risk of the disease or can be a consequence of the disease [214]. A study by Stefańska et al. [214] noted that the group of people with primary or vocational education was significantly higher among schizophrenia patients. 

In connection with unemployment, schizophrenia, in particular, has a negative impact on eating and health habits in general. In highly developed countries, unhealthy processed foods are cheaper and more readily available, which implies that schizophrenia patients consume more of these types of foods [215,216]. Patients with schizophrenia who abuse psychoactive substances are more likely to exhibit characteristics that put them at risk for these addictions, such as lower levels of education, homelessness, and childhood behavioral problems. The unemployment and social isolation that accompany schizophrenia may increase the social value of a person with schizophrenia in a group of addicts, making addictions likely to be seen as “attractive” among schizophrenic patients [217]. 

### 6.2. Nicotinism

The prevalence of smoking among people with schizophrenia is estimated to be two- to four-times higher than in the general population (20–35%), and the number of cigarettes smoked per day is on average four cigarettes higher than in the general population [218,219]. A meta-analysis conducted by Jose de Leon et al. (2005) clearly showed an association between schizophrenia and current smoking. The association persisted after using controls for those with severe mental illness and controlling for other variables. In addition, compulsive smoking and high nicotine dependence were more common in smokers with schizophrenia compared to the general population. Patients with schizophrenia were more likely to have smoked cigarettes compared to the general population and compared to patients with severe mental illness [219].

Moreover, in a study by Jose de Leon et al. (2002), the risk of starting daily smoking was increased in patients with schizophrenia compared to the controls. The study found that the risk of starting smoking before the age of 20 years was similar in both schizophrenia patients and controls; however, after the age of 20 years, the rate of starting daily smoking in schizophrenia patients was significantly higher [220].

There is also a hypothesis regarding the self-medication of smoking patients with schizophrenia. An analysis by Kumari et al. indicates that a percentage of patients reported an improvement in psychiatric symptoms, which some said worsened after quitting smoking. Moreover, such patients prefer stronger cigarettes and extract more nicotine from them than other smokers. The self-medication hypothesis itself suggests that nicotine has the effect of reducing the side effects of antipsychotics, enhancing the therapeutic effect of antipsychotics, thereby alleviating negative symptoms and/or alleviating a number of cognitive deficits associated with schizophrenia [218]. 

Given the higher prevalence of smoking among schizophrenics, it is of interest what the biological basis for this mechanism is.

Levin et al. conducted a study to understand the neurobehavioral basis of nicotine involvement in cognitive function [221]. Nicotine therapy targeting specific receptor subtypes and nicotine therapy with drugs that affect interacting transmitter systems appeared to have cognitive benefits precisely for syndromes characterized by disorders such as Alzheimer’s disease, schizophrenia, and attention deficit hyperactivity disorder. Numerous studies point to a link between neuronal acetylcholine receptors (nAChRs) and the attention processes and disorders observed in schizophrenia patients.

Postmortem examinations have revealed a decreased density of neuronal nicotinic receptors (NNRs), particularly the alpha7 subtype, in the hippocampus of schizophrenic patients [222]. Alpha7 NNR receptors are known to influence several key neurotransmitters implicated in schizophrenia, such as glutamate, GABA, and dopamine [223]. Agonists targeting alpha7 NNRs have shown promising results in reducing both positive and negative symptoms of schizophrenia and may also enhance cognitive function [224]. These observations indicate that alpha7 NNRs play a significant role in the manifestation of schizophrenia symptoms and highlight their potential as targets for developing comprehensive treatments for the disorder. This reduction in alpha7 NNRs might also explain why so many individuals with schizophrenia smoke cigarettes, as it could be an indirect indication of a self-medication attempt.

Researchers Jose de Leon and Francisco J. Diaz analyzed data available from studies conducted in five countries, which indicate that the percentage of people who intend to quit smoking is significantly lower among patients with schizophrenia and the general population. The denominator was the smoking cessation rate, which was 9% in patients and 19 to 49% in the general population [219]. Schizophrenia patients who have tried to quit smoking have reported ineffective social promotions of abstinence in the long term. Nevertheless, both pharmacological and psychosocial approaches are useful for schizophrenic patients in attempting to reduce and quit smoking in the short term [225]. Despite the improved well-being of schizophrenic patients who smoke, further attempts to quit are advisable due to the significantly associated health benefits, and future interventions to help patients to quit should focus on translating short-term benefits into long-term effects.

### 6.3. Alcohol

In schizophrenia patients, the second most common addiction, after nicotine, is alcohol. It has a dramatic effect on the frequency and intensity of psychotic episodes and life expectancy [226]. The reasons for the increased sensitivity to alcohol in patients with schizophrenia can be traced to damage to the reward center in the brain [227]. 

Data suggest that the pathophysiology and pharmacotherapy of schizophrenia contribute to an excess susceptibility to alcoholism and substance abuse compared to the general population. Moreover, schizophrenic patients whose first-degree relatives meet the diagnostic criteria for alcohol and substance abuse disorders have a higher risk of substance abuse than other schizophrenic patients. Additional data indicate that there is no observable increased incidence of schizophrenia in children of alcohol-dependent parents, while the incidence of alcohol dependence, substance abuse, or antisocial behavior in family members is not increased in schizophrenic patients who do not have comorbid substance abuse disorders [217].

C D’Souza et al. (2006) conducted a study to determine the dose-dependent effect of alcohol on symptoms and side effects of drugs in schizophrenic patients. They compared the effects of alcohol on schizophrenic patients and healthy individuals. The study concluded that schizophrenic patients experienced greater euphoria and agitation than healthy subjects. Alcohol did not affect negative symptoms in these patients, but only caused a temporary increase in positive symptoms and changes in perception [228]. Some authors have suggested the “hypothesis” of self-medication in schizophrenia, which involves the use of alcohol and psychoactive substances in schizophrenia patients [229]. According to this hypothesis, addictive substances help such individuals alleviate painful effects, experiences, or control their emotions. The above study by D’Souza et al. (2006) [228] does not support the “self-medication" hypothesis—alcohol did not reduce any of the underlying symptoms of the disease.

### 6.4. Number of Meals

A study conducted by Stefańska et al. found that women diagnosed with schizophrenia consumed a significantly higher number of meals per day compared to a group of healthy women. In addition, their breakfasts had an energy value that was too low compared to the recommendations, while the other meals were characterized by an energy value in line with the recommendations. There was a significant contribution of snacking to the provision of energy in the diet of women with schizophrenia (8%). Among female patients, sweets were the most frequently chosen snacks, whereas men tended to opt for candy or sandwiches [214]. These data are consistent with research by Kampov-Polevoy et al. (2006) [230], where women’s instinctive response to sweet taste is associated with increased sensitivity to the mood-altering effects of sweet foods and impaired control over eating sweets. This also coincides with a study by Stefańska et al. [214], which noted that the amount of carbohydrates consumed was significantly higher in a group of women with schizophrenia compared to healthy women. In contrast, the opposite trend was noted in the male group. In the group of men, no significant difference was observed in the number of meals consumed, with 3–4 meals being consumed per day in both groups. The energy value of lunches and afternoon snacks was also observed to be too low, while evening meals had too high an energy value compared to the recommendations. Interestingly, in the control group of men, the energy value of all meals during the day was found to be too low, except for dinner, which in turn showed too high an energy value.

## 7. Metabolic Impact of Antipsychotics on Schizophrenia Patients

Schizophrenia patients are significantly more likely to have diabetes, and the antipsychotic drugs they take further increase the risk of insulin-dependent hyperglycemia [231,232]. There are several hypotheses regarding glucose regulation in schizophrenic patients taking neuroleptics. One of them talks about the acquisition of an increased tolerance to insulin, which increases with the duration of the disease. Another hypothesis states that atypical antipsychotics affect the cellular insulin receptor directly by altering its binding properties [232]. 

### 7.1. Glucose Metabolism and Neuroleptics Treatment 

Of the atypical antipsychotics, clozapine and olanzapine are associated with the highest risk of weight gain, type 2 diabetes, and dyslipidemia. In contrast, risperidone and quetiapine can cause intermediate weight gain and varied metabolic changes, but additional results were inconclusive on the incidence of diabetes and dyslipidemia. Aripiprazole and ziprasidone showed the least weight gain, and there is no evidence of a risk of diabetes or dyslipidemia during or after their use [231]. It is worth noting that a study by Casey et al. showed that changing the treatment of patients previously taking olanzapine or risperidone to aripiprazole or ziprasidone resulted in a significant reduction in weight over 6 to 8 weeks compared to the weight of patients before treatment [233]. We have presented the data on this issue in Table 3.

The pathomechanisms involved in the occurrence of metabolic diseases are not fully understood, but it is assumed that weight gain occurs secondary to a central increase in appetite. This theory stems from the fact that histamine can inhibit the mesolimbic dopamine pathway, which controls the intake of palatable food via the H3 autoreceptor while activating it via the histamine H1 receptor [234]. A study by Kroeze et al. showed that the binding affinity of atypical antipsychotics to the histamine H1 receptor is a relevant predictor of weight gain mediated by a group of atypical antipsychotic drugs [235].

The results of the study indicate that olanzapine, which is associated with weight gain, causes a significant increase in postprandial insulin, glucagon-like peptide 1 (GLP-1), and glucagon levels, which co-occurs with insulin resistance compared to placebo. In contrast, aripiprazole, considered less metabolically damaging, induces insulin resistance, but has no effect on postprandial hormones. In addition, atypical antipsychotics, here olanzapine and aripiprazole, have been shown to directly affect tissues, i.e., postprandial glucose and hormone levels, as well as glucose metabolism and insulin resistance. The changes occur regardless of dietary mechanisms, as metabolic changes were seen in patients who showed no weight gain or increased food intake [236].

### 7.2. The Risk of Hiperlipidemia and Neuroleptic Treatment

There is growing clinical evidence that some antipsychotic drugs may increase the risk of lipid metabolism disorders. A number of case reports have noted an association between the use of clozapine and olanzapine and an increased risk of hypertriglyceridemia, an increase in serum total cholesterol, and hyperlipidemia that resolves when the drugs are discontinued [237]. Other studies show that after 5 years of clozapine treatment, average triglyceride levels double and cholesterol levels rise by at least 10% [3]. In a study, in an effort to test whether antipsychotics have acute effects on lipid levels and other metabolic profiles, mice were given an intraperitoneal injection of clozapine. Administered in this manner, clozapine rapidly induces direct transcriptional effects on the liver by influencing genes that control transcription factors, such as protein transcription factors binding sterol regulatory elements, peroxisome proliferator-activated receptors, and hepatic X receptors. This facilitated lipid deposition in the liver by increasing lipogenesis, as these genes are involved in fatty acid biosynthesis, regardless of the type of food consumed and weight gain. This treatment caused a significant increase in triglycerides, phospholipids, and cholesterol levels within 48 h in mice. According to a study by Olfson et al. (2006), first-generation antipsychotics and each of the second-generation antipsychotics (clozapine, risperidone, quetiapine, olanzapine, and zyprazidone), with the exception of aripiprazole, posed a risk of hyperlipidemia greater than not taking the drugs [237].

## 8. Conclusions

Growing evidence suggests that the gut–brain axis is crucial for understanding schizophrenia pathogenesis, linking it to neurodevelopmental, immunological, and metabolic pathways.

Further research is essential to understand the functioning of the gut–brain axis in schizophrenia patients. The challenge of understanding such correlations lies in the multitude of variables that affect gut health and, consequently, the axis. This complexity is exacerbated in individuals with schizophrenia due to the various social and environmental factors discussed in the main body of this work. To more definitively assess the role of the gut–brain axis, large-scale studies are needed to compare patients independently of environmental factors and diet. In this context, a long-term observation could also be beneficial. 

Data from studies conducted on patients using atypical antipsychotics indicate that they should undergo regular monitoring of weight, plasma glucose levels, and plasma lipid levels. Such actions would enable individual treatment decisions to be made and reduce the iatrogenic impact of drug use on morbidity and mortality, which is essential because, compared with the general population, persons with schizophrenia have up to a 20% shorter lifespan.

While traditional medications for schizophrenia may not always be effective, supplementary treatments may offer potential benefits. In the context of supplementation, guidelines from The World Federation of Societies of Biological Psychiatry (WFSBP) and Canadian Network for Mood and Anxiety Treatments (CANMAT) taskforce indicate that compounds, such as N-acetylcysteine, folate, and Ginkgo biloba, are more effective. Again, consistent with the data from our review, vitamin D at 1500 IU and 4000 IU is also ineffective. However, it is important to emphasize that, despite the significant limits, such doses of vitamin D may be too low to lead to a clinical improvement. Only well-designed studies on high doses of vitamin D can indicate its role in supplementation in schizophrenics. On the other hand, omega acids, despite also being irrelevant in the course of the disease, may be useful because of their ability to counteract the side effects of neuroleptic drugs, especially those that significantly contribute to the development of obesity. 

In conclusion, this review highlights the need for a holistic approach to the treatment of schizophrenia, including pharmacotherapy, nutritional supplementation, and modulation of the gut microbiota. The findings highlight the complexity of neural, endocrine, and immune interactions, and suggest that future research should focus on better understanding these mechanisms and their practical application in patient therapy. There may be significant clinical benefits to be gained from therapies supported by supplements such as omega-3 fatty acids, B vitamins, probiotics, and innovative drugs, such as GLP-1 receptor agonists. At the same time, to develop more effective therapeutic strategies, continued research into the impacts of diet and gut microbiota on mental health is critical.

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
