# Peer review of "Investigating the Impacts of Diet, Supplementation, Microbiota, Gut–Brain Axis on Schizophrenia: A Narrative Review"

_nutrients, 2024, doi:10.3390/nu16142228_

Round 1

Reviewer 1 Report

Comments and Suggestions for Authors

The manuscript reports an interesting narrative review of the relationships between schizophrenia and microbiota. The paper is well-written and clear. The data reported are balanced and there are no specific concerns. 

I have only a few comments:

- the introduction should end with a motivation for this narrative review.

- at the end of the introduction I would include a brief description of the points that you evaluated in the text to help the readers

- is it possible, in the conclusion, to propose an evaluation of the current literature - for example, if there are indications of areas with poor data? 

- it is not clear to me why you referred to schizophrenia as a neurological disease (like the subtitle 4.1). This aspect should be revised. It is a psychiatric disorder.

Author Response

Dear Reviewer,

Thank you for your work. I am pleased that the paper is balanced and clear. I will now address your comments:

  • The introduction should end with a motivation for this narrative review: Thank you for this comment. We have added a specific objective and motivation for our review.

  • At the end of the introduction, I would include a brief description of the points that you evaluated in the text to help the readers: Of course, we have added bullet points outlining the topics we will discuss.

  • Is it possible, in the conclusion, to propose an evaluation of the current literature - for example, if there are indications of areas with poor data?: Thank you for this suggestion. Indeed, our conclusion was imprecise. In response, we have revised it to provide more specific opinions and highlight areas that require further research. Please see the attachment.

  • It is not clear to me why you referred to schizophrenia as a neurological disease (like the subtitle 4.1). This aspect should be revised. It is a psychiatric disorder: Thank you, we have corrected this and searched the paper for similar errors. We have renamed one mismatched section and edited minor mistakes.

Best regards,

Stefan Modzelewski

Reviewer 2 Report

Comments and Suggestions for Authors

The paper is actual, particularly important not only from theoretical point of view, but also for clinical practice.

Gut-brain axis is a hot topic . Supplements and microbiota-targeted interventions for mental health may improve the symptoms of serious mental illnesses which are often pharmacoresistant and chronic. Further,  add-on strategies known to affect the gut-microbiome for the treatment of schizophrenia are important as for availability, and patientsʹ preference.

I have only one suggestion - please include  ad subchapter five the following:

Sarris J, et al. Clinician guidelines for the treatment of psychiatric disorders with nutraceuticals and phytoceuticals: The World Federation of Societies of Biological Psychiatry (WFSBP) and Canadian Network for Mood and Anxiety Treatments (CANMAT) Taskforce. World J Biol Psychiatry. 2022 Jul;23(6):424-455. These guidelines are the only one dealing in a comprehensive a detailed way this topic.

Author Response

Dear Reviewer,

Thank you for your work. We are glad that our work fits into current trends.

I will try to respond to your suggestion:

''I have only one suggestion - please include  ad subchapter five the following:

Sarris J, et al. Clinician guidelines for the treatment of psychiatric disorders with nutraceuticals and phytoceuticals: The World Federation of Societies of Biological Psychiatry (WFSBP) and Canadian Network for Mood and Anxiety Treatments (CANMAT) Taskforce. World J Biol Psychiatry. 2022 Jul;23(6):424-455. These guidelines are the only one dealing in a comprehensive a detailed way this topic.'' --> Thank you for this valuable comment. Of course, We have read this work and added it to our review. Please see the attachment.

Best regards,

Stefan Modzelewski
